# Examples of Underexploited Marine Organisms in Cosmeceutical Applications

**DOI:** 10.3390/md23080305

**Published:** 2025-07-30

**Authors:** Céline Couteau, Laurence Coiffard

**Affiliations:** Laboratoire interactions Epitheliums Neurones (LIEN), UR 4685, Université de Bretagne Occidentale, 22 Avenue Camille Desmoulins, 29200 Brest, France; celine.couteau@univ-nantes.fr

**Keywords:** cosmetic products, marine resources, underexploitation

## Abstract

A number of marine resources have been exploited for a long time. Examples include *Fucus* and *Laminaria*, from which gelling agents are extracted. Only a few hundred thousand marine species are currently known, representing a tiny fraction of the estimated total of between 700,000 and one million species. This opens up numerous possibilities for innovation in the cosmetics industry. In this study, we present various species that are currently under-exploited, but which could have applications in hydration and photoprotection, for example. Algae and microalgae are worthy of interest because they can be used for hydration and anti-ageing purposes. Collagen can be extracted from animal sources and used as a substitute for collagen of bovine origin. From a marketing perspective, it is possible to market it as ‘marine collagen’. However, it is imperative to emphasize the significance of ensuring the sustainability of the resource. In accordance with this imperative, algae that are capable of being cultivated are distinguished by their enhanced qualities.

## 1. Introduction

The sea has always fascinated humanity, while also inspiring terror, as it can be both nourishing and deadly depending on the circumstances. Since ancient times, it has been a prevalent theme in literature, as evidenced by Noah’s journey on the raging waves [1] and Ulysses’s endless maritime voyage, during which he faced a myriad of perils, including the encounter with the fearsome sea monsters Charybdis and Scylla [2]. The movement of the waves has subsequently inspired many poets and novelists. Descriptions of swells reflecting inner storms and exaltations of immensity permeate the verses of Du Bellay, Baudelaire and Tristan Corbière [3]. For Bernardin de Saint-Pierre and Jules Verne, the sea is a sign of adventure [4,5]. The marine world has also inspired numerous pictorial works.

We can see the sea’s strong presence in art, and it also occupies a central place in everyday life, particularly in food, as is the case in Japan and certain European countries such as Spain, Portugal, Greece, and Norway [6]. Since Antiquity, the sea has also played a significant role in skincare [7,8]. This explains why some consumers, attracted by both the sea and ingredients of natural origin, demand marine cosmetics. Specialized structures have been developed for the supply of marine raw materials. They most often offer liquid extracts in a mixture of water and propylene or butylene glycol. Key players in this sector include Ceva (Pleubian, Côtes d’Armor, France), Codif International (Ille-et-Vilaine, France), Secma (Pontrieux, Côtes d’Armor, France), and Gelyma (Bouches-du-Rhône, France). It is evident that prominent brands, including Phytomer, Thalgo, Algologie, and Algotherm, have strategically aligned their marketing endeavors with the notion that marine raw materials possess inherent benefits for the skin. These marks correspond to the «blue cosmetics». It is evident that a significant number of other brands, albeit less frequently, extend their marketing efforts to the marine sector in order to promote a particular product. Here, we will present ingredients of interest that are not widely used yet, which could help particular cosmetic brands to meet consumer expectations. Algae have long been used to produce additives such as alginates and carrageenans [9,10], as well as being used as active ingredients in the form of *Fucus vesiculosus* extracts in slimming cosmetics [11]. Here, we present substances of marine origin obtained from algae or marine invertebrates that are still underused. These substances have applications in the fields of hydration and anti-ageing, as well as skin barrier function and topical photoprotection.

## 2. Underexploited Marine Ingredients That Could Be of Interest to the Cosmetics Industry

### 2.1. Algae

#### 2.1.1. Moisturizing Effect

The water content of the skin, particularly in the *Stratum corneum* (the outermost layer of the epidermis), is essential for maintaining skin hydration. A value below 10% can result in dry and uncomfortable skin, which is sometimes accompanied by itchiness, particularly in elderly individuals [12,13], and can lead to a loss of qualities such as softness and suppleness. In the stratum corneum, water binds to hygroscopic substances collectively known as the Natural Moisturizing Factor (NMF). The NMF is mainly composed of amino acids (including serine), pyrrolidone carboxylic acid, lactic acid, urea, sugars, and minerals. Keratinization plays an important role in forming the molecules that make up the NMF. The binding of water to NMF compounds constitutes the static aspect of skin hydration. The second dynamic aspect relates to the *Stratum corneum*’s selective permeability and lipid barrier properties. The degree of permeability depends on the integrity of the *Stratum corneum*, as well as on the nature and organization of the intercorneocyte lipids [14,15]. Therefore, the formulation of moisturizing products is based on two strategies: hydrophilic substances that can retain water in the stratum corneum, such as humectants like glycerin and glycols, and true moisturizers like urea and lactic acid; and lipophilic, occlusive and film-former ingredients that can oppose excessive transepidermal water loss (TEWL) and protect the skin against certain external aggressions [16]. In this context, polysaccharides extracted from algae may provide an alternative to glycols, which are sometimes criticized. The chemical structure and types of polysaccharides found in marine algae differ considerably from those found in terrestrial plants, reflecting the particular environmental conditions in which these organisms develop. Their moisturizing effect is likely to be due to the three-dimensional structure resulting from the configuration of the polysaccharide molecular chains [17,18]. This dense network allows water to be retained in the skin due to its film-forming properties. Furthermore, the polar groups of the polysaccharide chains are likely to form hydrogen bonds, a phenomenon which promotes skin hydration when coupled with the dense network structure [19]. In this respect, enteromorphs appear to be good candidates. The green alga *Enteromorpha prolifera*, for example, constitutes an important biomass resource due to its rapid proliferation in the intertidal zone or eutrophic coastal waters. The polysaccharide of *E. prolifera* consists mainly of rhamnopyranosyl (1 → 4) and (1 → 2, 4) residues, as well as xylopyranosyl (1 → 4) and glucuronosyl (1 → 4) residues, along with terminal residues. The sulphate groups are primarily found at C-3 of the (1 → 4) rhamnose residues and at C-2 of the (1 → 4) xylose residues. This polysaccharide exhibits an interesting moisture retention capacity due to its high sulfate content [18]. Another green alga of interest is *Ulva fasciata*, as the sulfated polysaccharide extracted from it has been shown to be a more effective humectant than glycerol, the traditional reference [20]. There are two mechanisms that can be implemented to ensure the stratum corneum is well hydrated: using a film-forming ingredient or a hydrophilic substance that retains water in situ (see Figure 1).

These two mechanisms are most often used simultaneously in a formula using 2 types of ingredients. In the particular case of polysaccharides, both mechanisms are provided simultaneously.

#### 2.1.2. Anti-Wrinkle/Anti-Ageing Activity

It is now well established that oxidative stress caused by reactive oxygen species (ROS) plays a significant role in the ageing process in general, and in skin ageing in particular. EROs are often generated uncontrollably under pathological conditions or due to external factors and agents [21]. Therefore, the main active ingredients in anti-ageing cosmetics are antioxidants such as vitamins A, C, and E, plant extracts rich in polyphenolic substances such as *Ginkgo biloba*, and hyaluronic acid [22,23]. While the vast majority of these ingredients are safe, some people may experience adverse reactions. One example is retinol, which makes the skin more sensitive to ultraviolet radiation and should therefore not be applied to sensitive skin. The use of algae extracts can be considered as an effective substitute. *Furcellaria lumbricalis* is a red alga that is mainly used for producing carrageenan [24]. Other ways of exploiting this alga should be considered, as it could provide an interesting source of phycoerythrin [25]. We will be interested in its potential anti-cellulite and anti-ageing activity. When associated with *Fucus vesiculosus* and retinol, this alga could promote lipolysis and stimulate procollagen I synthesis [26]. These properties have been demonstrated on cultured adipocytes and fibroblasts, respectively. This could lead to applications in the field of slimming products and anti-ageing cosmetics. Additional studies are necessary because the results of these studies make it impossible to conclude what share *Furcellaria lumbricalis* has in the observed effect. Furthermore, it has been demonstrated that an extract from the brown alga *Alaria esculenta* is capable of significantly reducing the amount of progerin in cultured fibroblasts [27]. Progerin is a truncated version of lamin A that acts with telomeres to trigger cellular senescence in human fibroblasts. Over time, as the skin ages, progerine accumulates and is expressed more in older fibroblasts than in younger ones [28]. Photoaging, also known as solar elastosis, is a component of extrinsic aging and a complex, multifactorial process triggered by UV irradiation. It induces oxidative stress and positively regulates the expression of matrix metalloproteinases. It also degrades the extracellular matrix (ECM) and causes an inflammatory response. These processes result in the characteristic signs of photoaging, such as the rhomboid nape [29]. In UVB-irradiated human HaCaT cells, the polysaccharides of *Hizikia fusiformis* (also known as *Sargassum fusiforme*) have been shown to significantly improve cell viability and hydroxyproline secretion. In particular, the extract obtained by UV/H_2_O_2_ treatment for 45 min exhibited the most effective anti-photoaging properties. Furthermore, this extract was found to significantly increase collagen I content and expression, while decreasing the levels of pro-inflammatory cytokines, including interleukin-1β, interleukin-6, and tumor necrosis factor-α [30]. In addition to wrinkles and fine lines, skin ageing is manifested by skin spots known as senile lentigines [31]. The cosmetic industry markets lightening agents to eliminate or at least reduce their visibility [32]. However, there is a shortage of research in this area. Hydroquinone has proven to be effective, but it is now banned due to its toxicity [33]. Although kojic acid has also been shown to be effective as a skin brightener, some authors claim that it is mutagenic [34]. Therefore, the availability of new illuminating ingredients is urgent. In 2005, a patent was filed for a cosmetic composition containing a mixture of floridoside, extracted from *Dilsea carnosa*, and isethionic acid as active substances [35]. This mixture is proposed not only as a skin brightener and inhibitor of melanogenesis, but also for skin hydration and the prevention of skin ageing.

#### 2.1.3. Action on the Skin Barrier Function

*Ishige okamurae* is one of the most common edible brown algae. Widely distributed in the coastal areas of East Asia, it is characterized by its high polyphenol content and presence of fucoxanthins and phlorotannins, such as diphlorethohydroxycarmalol (Figure 2), as well as ishigosides.

The antioxidant and anti-inflammatory properties of this alga have been demonstrated [36]. Its action on the skin’s barrier effect is more original. Indeed, it has been confirmed that diphlorethohydroxycarmalol increases the expression of skin barrier proteins (lymphopoietin, stromal thymus, filaggrin, loricrin, and involucrin) in an atopic dermatitis model induced by DNCB, as well as in HaCaT cells. In addition, the expression of tight-junction proteins (such as claudins, occludins, and tight-junction protein 1) improved [37]. Such extracts could be of great interest when formulating cosmetic products for atopic skin. Indeed, while the primary treatment for eczema is the application of dermocorticoids, it is now common practice to use emollients (i.e., cosmetic products) between flare-ups [38]. Any ingredient that promotes the proper functioning of the skin barrier will therefore have a positive effect on the patient.

#### 2.1.4. Photoprotective Effect

In the field of photoprotection, the European Union currently permits around thirty UV filters (see Table 1).

The total number of filters available for use in solar products is, therefore, limited, as is the number of UVA filters. Furthermore, many of these filters are controversial, being classified as endocrine disruptors [39] and/or sensitizers [40], as well as emerging pollutants [41]. Therefore, active research into synthesizing new, safer filters and valorizing extracts obtained from superior plants or algae is crucial. Work in this area is extensive. Here, we cite examples of algae that warrant further investigation given the current lack of UVA filters. For many years, studies have focused on low molecular weight, water-soluble mycosporin-like amino acids (MAAs) (Table 2), which have a maximum absorption range of 310–360 nm.

Four MAAs were identified in *Gracilaria vermiculophylla*: Porphyra-334, shinorine, palythine, and asterina-330. The highest levels of Porphyra-334 and shinorine were found between November and January, and the highest levels of palythine and asterina-330 were found between April and August. The main issue with exploiting them is the very small quantity produced: 8.56 g of MAAs in an 18 m^2^ crop over eight months, or a total of 71.33 g per year with an optimized system [42]. These quantities are extremely small when considered alongside the tons of UV filters required each year. It would also be necessary to determine the SPF of creams formulated using a particular derivative or combination of MAAs. Indeed, the SPF—the index proposed by Rudolf Schulze in the 1950s—remains the almost universal index for quantifying the efficacy of a preparation today [43].

### 2.2. Microalgae

Several microalgae species have been available as ingredients for around 20 years (see Table 3).

As before, we have chosen to present microalgae of interest that are not well known.

The anti-ageing and skin barrier-strengthening potential of Blue Lagoon coccoid filamentous microalgae extracts has been evaluated in vitro and in vivo [44]. These extracts have demonstrated the ability to increase the transcriptional expression of genes such as involucrin, loricrin, transglutaminase-1, and filaggrin, which are major markers of skin barrier function [45]. Furthermore, UV radiation accelerates collagen degradation by increasing metalloproteinase-1 (MMP-1) expression in fibroblasts. Blue Lagoon extracts are also likely to suppress MMP-1 hyperregulation and UVA-stimulated type 1 procollagen deregulation. Topical treatment with microalgae extracts incorporated into preparations at concentrations between 0.25% and 2.5% was found to reduce insensitive water loss, thus improving skin hydration. Furthermore, these extracts were found to improve the skin’s barrier function and demonstrate a preventive capacity for premature skin ageing [46]. Peptide fractions (430–1350 Da) isolated from *Chlorella pyrenoidosa* were shown to inhibit MMP-1 expression in irradiated fibroblasts in the UVB domain by suppressing AP-1 production, CYR61, and MCP [47]. Topical treatment with microalgae extracts incorporated into preparations at concentrations between 0.25% and 2.5% was found to reduce insensitive water loss, thus improving skin hydration. Furthermore, these extracts were found to improve the skin’s barrier function and demonstrate a preventive capacity for premature skin ageing [46]. Peptide fractions (430–1350 Da) isolated from *Chlorella pyrenoidosa* were shown to inhibit MMP-1 expression in irradiated fibroblasts in the UVB domain by suppressing AP-production, CYR61, and MCP [47]. Topical treatment with microalgae extracts incorporated into preparations at concentrations between 0.25% and 2.5% was found to reduce insensitive water loss, thus improving skin hydration. Furthermore, these extracts were found to improve the skin’s barrier function and demonstrate a preventive capacity for premature skin ageing [46]. Peptide fractions (430–1350 Da) isolated from *Chlorella pyrenoidosa* were shown to inhibit MMP-1 expression in irradiated fibroblasts in the UVB domain by suppressing AP-production, CYR61, and MCP [47]. Some species of microalgae have adapted to UV radiation, making them of interest. However, the molecular mechanisms underlying their response to stress, adaptation, and resilience to this radiation are not well understood. However, some of these mechanisms have been identified [47]. For example, the species *Eutreptiella* sp. from the Southern Ocean has successfully used photoprotective pigments to protect itself under variable light conditions. Similarly, the Antarctic microalgae *Chaetoceros dichotoma*, *Phaeocystis antarctica*, and *Polarella glacialis* exhibited particular resistance to UV-B radiation. In the case of *P. glacialis* in particular, this strain has been found to have a very high xanthophyll/chlorophyll ratio, indicating a high concentration of UV-absorbing compounds [47]. The strains *P. antarctica* and *P. glacialis* showed strong induction of UV-absorbing compounds when subjected to increasing intensities [48]. Studies have also shown that sporopollenin, a natural biopolymer resistant to acetolysis, present in microalgae, could be of interest for topical photoprotection. This material, of lipid nature, whose exact composition is not fully elucidated, was first discovered in the outer wall of the spores, then in the shell of the pollen. Finally, this biomaterial was found in the walls of unicellular algae such as *Chlorella vulgaris* [49,50]. As with macroalgae, the vital role of different microalgal antioxidants (MAAs) has been demonstrated in the protective mechanisms of several microalgal strains, due to their capacity to absorb UV radiation and their antioxidant properties [51]. Some authors consider MAAs to be the most effective agents for protecting microalgal genetic material due to their properties [50]. For this reason, they could potentially prevent UV-induced skin damage [51,52]. Peptide fractions obtained from *Chlorella pyrenoidosa* have been shown to inhibit the expression of MMP-1 induced by UVB in cultured fibroblasts by suppressing the production of AP-1, CYR61, and MCP [53]. This suggests potential applications in the anti-ageing cosmetics industry. To conclude this overview of interesting microalgae strains, let’s look at *Botryococcus braunii*. This microalgae strain has been shown to stimulate adipocyte differentiation in a dose-dependent manner. An aqueous extract at a concentration of 0.1% was found to be more effective than nicotinamide, a well-known activator of adipocyte differentiation. The mechanism of action involves the inhibition of cAMP induced by certain substances, such as forskolin. The same extract was also found to increase the expression of aquaporin-3, filaggrin, and involucrin in an in vitro model [54]. This is interesting given that aquaporin 3 (AQP3) facilitates the formation of an osmotic gradient between the different layers of skin, thereby promoting the hydration of the stratum corneum [55]. The significant increase in aquaporin-3 in keratinocytes treated with an aqueous extract of *Botryococcus braunii* can be explained by the translocation of aquaporin-3 from the cytoplasm to the plasma membrane. Numerous studies have now demonstrated the pivotal function of aquaporins in maintaining skin hydration [56,57]. Additionally, results similar to those obtained with vitamin C were found in terms of stimulating collagen synthesis [54]. Finally, antioxidant and anti-inflammatory activity was also observed [54]. These results could lead to the cosmetic industry using these microalgae for multiple applications. Microalgae produce squalene, a widely used cosmetic ingredient with emollient properties. An oxidized form of squalane, squalene is a biologically active terpenoid produced by animals and plants. Currently, sharks are the main source of squalene. About 3000 sharks must be killed to produce one ton of squalene. However, these animals are on the verge of extinction. This situation, therefore, provides a strong incentive to turn to alternative sources and strains of microalgae that produce squalene in a sustainable way. Such sources are of interest [58]. The genera *Schizochytrium*, *Aurantiochytrium*, and *Thraustochytrium* appear to be particularly promising [59,60].

### 2.3. Underexploited Extracts of Marine Animals

Animal-derived raw materials are certainly not the most popular choice for some consumers. However, some raw materials can be obtained from vertebrates or marine invertebrates. One example is collagen, which can be extracted from fish skin as an alternative to bovine or porcine collagen and is then described as marine collagen [61,62]. As well as being rejected by some consumers, marine collagen has the advantage of being less likely to cause health problems, such as the transmission of bovine spongiform encephalopathy, transmissible spongiform encephalopathies, and foot-and-mouth disease. Various techniques can be employed to recover fish waste [63]. Collagen has long been used in anti-ageing cosmetics due to its important role in skin firmness [64]. An in vivo study demonstrated the efficacy of collagen derived from blue shark cartilage when incorporated into a gel at concentrations ranging from 0.125% to 5% and applied to the skin on the wrists of volunteers. Twenty minutes later, the hydration level, skin texture, complexion, and sebum level were assessed. The results suggest that adding the extract improves all the evaluated parameters and has a wrinkle-smoothing effect [64]. Another study investigated the moisturizing effect of halibut skin collagen, which was also incorporated into a hydrogel [65]. *Tilapia* fish also appear to be a promising source of collagen [66]. In various cosmetic applications, ovothiols—sulfur metabolites produced by several marine invertebrates, such as the sea urchin *Paracentrotus lividus*—would be quite relevant to consider. Their exceptional antioxidant properties are linked to the position of the sulfhydryl group on the imidazole ring of histidine [67]. Potential sources of collagen include sea cucumbers, mussels, sea anemones, shrimp, starfish, jellyfish, sponges, sea urchins and squid, and octopus [68]. Sea cucumbers are interesting due to their many bioactive compounds, including triterpene glycosides, glycosaminoglycans, gangliosides, collagen, and branched-chain fatty acids [69]. Extracts from *Holothuria atra* and several other species of sea cucumber, including *H. arguinensis* from the north-east Atlantic, were found to have high antioxidant potential [70,71]. Anti-ageing products can benefit from this. In traditional Asian medicine, sea cucumber has long been used to treat wounds. Protein derivatives (typically 1.0 kDa) have been tested for their wound-healing ability in diabetic animal models. Thanks to their anti-inflammatory, antioxidant, and angiogenesis-inducing properties, these derivatives could help to speed up the healing process [72]. The mussels are also worthy of interest. A skincare preparation containing glycogen derived from *Mytilus coruscus* helps to control the complications of certain inflammatory skin conditions and prevents the effects of chapped skin and skin ageing [73]. A preparation containing an active peptide isolated from *Mytilus coruscus* is also an excellent anti-inflammatory agent [74]. An enzymatic hydrolysate of *Apostichopus japonicus*, containing peptides with a molecular weight of less than 3 kDa and other amino acids, has been shown to exhibit anti-ageing properties by increasing the expression of the Klotho protein and inhibiting lipid peroxidation [75]. Although jellyfish are often considered pests, they are of interest to the cosmetics industry as a source of collagen and fatty acids. For example, the genera *Chrysaora*, *Cyanea*, and *Stomolophus* have a higher polyunsaturated fatty acid (PUFA) content than other scyphomedusae [76]. Despite their potential, marine invertebrates are clearly a category of animals that is too underexploited.

The ensuing discourse will center on the topic of MAAs. It is noteworthy that the accumulation of these substances within vertebrates, including fish species, serves as a salient exemplar of symbiosis. Fish lack the necessary biosynthesis pathways, so they accumulate MAAs through their algal diet or via bacterial or algal symbionts. In addition to their food sources, de novo synthesis of gadusol (the MAAs’ precursor) has been observed in corals and fish [77,78,79]. The *Pocillopora capitate* coral species has a wide range of MAAs, particularly mono- and disubstituted ones such as mycosporin-Glycine, porphyra-334, shinorine, mycosporin-methylamine-serine, palythine-serine, mycosporin-methylamine-serinethreonine, palythine, and palythine-threonine [80].

A significant problem that should be given full consideration is the sustainability of the resource in question, given its exploitation of marine animals. It is evident that consumers are modifying their purchasing behaviors, with an increasing propensity to be attentive to natural products. Nevertheless, sustainability has been shown to be a pivotal factor in shaping intentions. Furthermore, a notable generational effect has been identified, younger demographics demonstrating a heightened level of concern for resource management [81].

## 3. Conclusions

Marine resources have barely been exploited and offer the cosmetics industry multiple sources of innovation. In the domain of hydration, for instance, it is intriguing to substitute conventional glycols with alternative ingredients that exhibit superior performance at reduced dosages. In the domain of photoprotection, there is an urgent need to replace certain UV filters, such as octocrylene or octylmethoxycinnamate, with molecules that do not possess any adverse effects. It must never be forgotten that there is a long road from the discovery of substances of interest to their marketing as raw materials in commercial products. The majority of the aforementioned substances are still in the experimental stage of development, with their effectiveness being demonstrated solely through in vitro testing. Many tests are necessary to ensure consumer safety. It is imperative to ascertain that the novel materials are non-toxic, non-irritant, and non-allergenic. Furthermore, as previously stated, the sustainability of the resource in question must be given due consideration.

## Figures and Tables

**Figure 1 marinedrugs-23-00305-f001:**
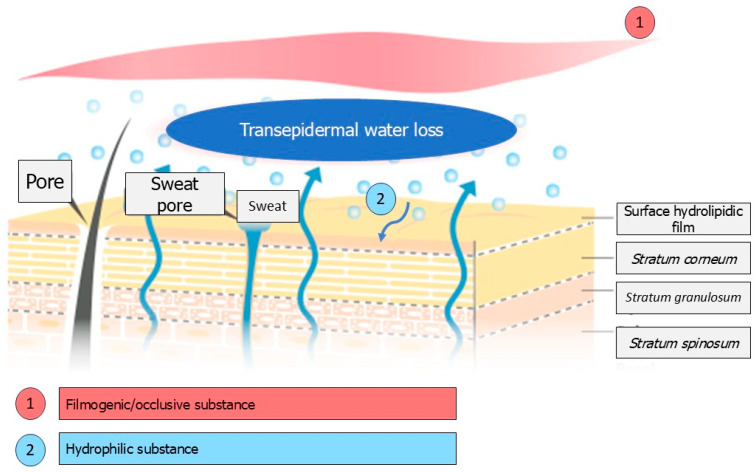
Mechanisms are implemented to ensure the *Stratum corneum* is hydrated.

**Figure 2 marinedrugs-23-00305-f002:**
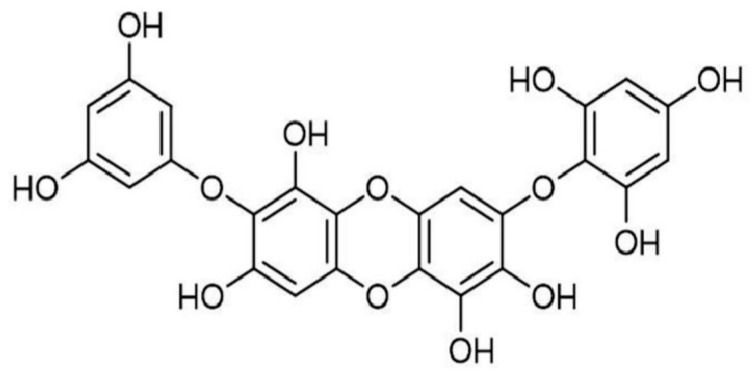
Chemical structure of diphlorethohydroxycarmalol.

**Table 1 marinedrugs-23-00305-t001:** List of UV filters permitted in cosmetic products in the European Union.

Camphor Benzalkonium Methosulfate
Homosalate
Benzophenone-3
Phenylbenzimidazole Sulfonic Acid
Terephthalylidene Dicamphor Sulfonic Acid
Butyl Methoxydibenzoylmethane
Benzylidene Camphor Sulfonic Acid
Octocrylene
Polyacrylamidomethyl Benzylidene Camphor
Ethylhexyl Methoxycinnamate
PEG-25 PABA
Isoamyl p-Methoxycinnamate
Ethylhexyl Triazone
Drometrizole Trisiloxane
Diethylhexyl Butamido Triazone
Ethylhexyl Salicylate
Ethylhexyl Dimethyl PABA
Benzophenone-4, Benzophenone-5
Methylene Bis-Benzotriazolyl Tetramethylbutylphenol
Methylene Bis-Benzotriazolyl Tetramethylbutylphenol [nano]
Disodium Phenyl Dibenzimidazole Tetrasulfonate
Bis-Ethylhexyloxyphenol Methoxyphenyl Triazine
Polysilicone-15
Titanium dioxide Titanium dioxide [nano]
Diethylamino Hydroxybenzoyl Hexyl Benzoate
Tris-biphenyl triazine Tris-biphenyl triazine [nano]
Zinc oxide Zinc oxide [nano]
Phenylene bis-diphenyltriazine
Methoxypropylamino Cyclohexenylidene Ethoxyethylcyanoacetate
*Bis*-(Diethylaminohydroxybenzoyl Benzoyl) Piperazine *Bis*-(Diethylaminohydroxybenzoyl Benzoyl) Piperazine [nano]

**Table 2 marinedrugs-23-00305-t002:** Main mycosporine-like amino acids.

MAAs	Maximal Wavelength (λ_max_) (nm)
Porphyra-334	334
Mycosporine-glycine	310
Shinorine	334
Palithynol	332
Palithene	360
Palithine	320

**Table 3 marinedrugs-23-00305-t003:** Examples of some commercially available microalgae extracts.

Microalgae	Trade Names	Suppliers	Composition of the Extract	Claims
*Chlorella vulgaris*	Dermosclupt	Codif	Glycerin Water Sodium benzoate Citric acid	Anti-ageing
*Chlorella vulgaris*	Dermochlorella D/DP	Codif	Water	Anti-angiogenic, anti-dark circle and refirming agent
*Dunaliella salina*	PEPHA-CTIVE	DSM	Water Phenoxyethanol Sodium benzoate Potassium sorbate	Hydration
*Dunaliella salina*	IBR-AAC	Lucas Meyer	Hydrogenated olydecene	Anti-ageing
*Dunaliella salina*	IBR-CLC	Lucas Meyer	Squalane Jojoba oil	Anti-ageing
*Duneliella salina* *Haematococcus vulgaris*	SUN’ALG	Gelyma	Karanja oil	Soothing
*Porphyridium cruentum*	ALGUARD	Frutarom	Phenoxyethanol	Anti-ageing
*Porphyridium cruentum*	SILIDINE	Greentech	Water Citric acid Sodium benzoate Potassium sorbate	Refirmind agent
*Phormidium persicinum*	Phormiskin Bioprotech	Codif	Glycerine Sea water	Anti-ageing, Radiance, mattifying, detoxifying
*Scenedesmus rubescens*	PEPHA-AGE	DSM	Water Phenoxyethanol	Anti-ageing

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
