# Peer review of "Examples of Underexploited Marine Organisms in Cosmeceutical Applications"

_marinedrugs, 2025, doi:10.3390/md23080305_

Round 1
Reviewer 1 Report
Comments and Suggestions for Authors
Comments
The manuscript entitled “Under-Exploited Marine Organisms” written by Céline Couteau and Laurence Coiffard summarized various species that are currently under-exploited with potential applications in hydration and photoprotection. This topic would be of interest among these researchers in marine organisms. This paper generally meets the requirement of this journal. However, there are some major issues in the paper that need to be addressed. Above all, I prefer to accept this manuscript for publication after addressing following points:
- The scope of the title is too large, which could not summarize the main ideas of the paper. The specific application was absent.
- The introduction of background in the abstract is too long, and the methods, results, and significance of this review are absent in the abstract.
- To our knowledge, there are too many bioactive ingredients in cosmetics industry from marine animals, such as peptides from shellfishes. Thus, the limited descriptions of part 2.3. “Under-Exploited Extracts of Marine Animals” without subtitles like part 2.1 were incomplete.
- The title of figure 2 should be revised as “Chemical structure of diphlorethohydroxycarmalol”. Besides, the chemical structures in ACS 1996 style of more related bioactive ingredients should be provided.
- All the latin names of marine organisms should be in italic. Such as, Line 69: prolifera; Line 102, Furcellaria lumbricalis; …Lines 103, 125, 205,
- Please check and revise all the refs.
Author Response
The scope of the title is too large, which could not summarize the main ideas of the paper. The specific application was absent.
1) The title has been modified.
The introduction of background in the abstract is too long, and the methods, results, and significance of this review are absent in the abstract.
2) The summary has been modified.
To our knowledge, there are too many bioactive ingredients in cosmetics industry from marine animals, such as peptides from shellfishes. Thus, the limited descriptions of part 2.3. “Under-Exploited Extracts of Marine Animals” without subtitles like part 2.1 were incomplete.
3) A comprehensive understanding of cosmetic ingredients reveals that there is a paucity of ingredients of marine origin.
The title of figure 2 should be revised as “Chemical structure of diphlorethohydroxycarmalol”. Besides, the chemical structures in ACS 1996 style of more related bioactive ingredients should be provided.
4) The title of Figure 2 has been rectified.
All the latin names of marine organisms should be in italic. Such as, Line 69: prolifera; Line 102, Furcellaria lumbricalis; …Lines 103, 125, 205,
5) It should be noted that all the names of the organisms mentioned are now printed in italics.
Please check and revise all the refs.
6) The bibliographic references have been meticulously reviewed.
Reviewer 2 Report
Comments and Suggestions for Authors
The manuscript from Couteau and Coiffard is a literature review describing potential cosmetics and bioactive compounds from under-exploited marine organisms.
Although the manuscript has merit in enlightening the scientific community on the potential exploit of less used marine organisms for new and effective bioactive compounds production and commercialization, there are several criticisms the authors should tackle before the publication.
The first point is that the title of the review is about “marine organisms” but then in the text, mainly micro and macroalgae potentialities are described, and very few information is given on other marine sources, just a very short paragraph dealing with sea urchins and corals. I don’t consider fish an under exploited source so I would not give so much space in the review. If the authors want to describe alternative sources for collagen there are many from marine invertebrates that have been described and they could include in the review. So, if the authors want to maintain the title they have to work on this part and give more space to other organisms. Indeed, several invertebrates come to mind such as Sponges, Echinoderms such as sea cucumbers, Cnidarians, just to cite some of them. The literature is rich of potential compounds with bioactive properties also useful for cosmetic purposes extracted and characterized from these organisms. There are so many possibilities that the authors did not consider.
Another important point is the Introduction: too short, mainly focused on classic literature and not the scientific one, not really describing the potential of exploitation of bioactive molecules from marine organisms. Furthermore, it does not describe state of the art of the most used marine organisms for cosmetic compounds.
An important issue that authors do not mention is that of sustainability. Is it really possible the explotation of these marine sources in a sustainable way, are there organisms fow which it would be easier to set up acquaculture facilities and other that would be difficult? Then there could be alternatives for their exploitment?
Finally the “Conclusion” paragraph is barely there. The authors should summarize better the results of their meta-analysis.
There are some mistakes here and there, so I would advise to revise the grammar and syntax.
Comments on the Quality of English Language
There are some mistakes here and there, so I would advise to revise the grammar and syntax. Some sentences are too colloquial e.g. "We are going to talk about MAAs here, as the accumulation of these substances...."
In table 3 "Acide citrique" is in french.
Author Response
The first point is that the title of the review is about “marine organisms” but then in the text, mainly micro and macroalgae potentialities are described, and very few information is given on other marine sources, just a very short paragraph dealing with sea urchins and corals. I don’t consider fish an under exploited source so I would not give so much space in the review. If the authors want to describe alternative sources for collagen there are many from marine invertebrates that have been described and they could include in the review. So, if the authors want to maintain the title they have to work on this part and give more space to other organisms. Indeed, several invertebrates come to mind such as Sponges, Echinoderms such as sea cucumbers, Cnidarians, just to cite some of them. The literature is rich of potential compounds with bioactive properties also useful for cosmetic purposes extracted and characterized from these organisms. There are so many possibilities that the authors did not consider.
1) The title has been modified.
Another important point is the Introduction: too short, mainly focused on classic literature and not the scientific one, not really describing the potential of exploitation of bioactive molecules from marine organisms. Furthermore, it does not describe state of the art of the most used marine organisms for cosmetic compounds.
2) The paragraph concerning marine animals has been expanded.
An important issue that authors do not mention is that of sustainability. Is it really possible the explotation of these marine sources in a sustainable way, are there organisms fow which it would be easier to set up acquaculture facilities and other that would be difficult? Then there could be alternatives for their exploitment?
3) The issue of the durability of the resource was addressed.
Finally the “Conclusion” paragraph is barely there. The authors should summarize better the results of their meta-analysis.
4) The conclusion has been expanded.
Reviewer 3 Report
Comments and Suggestions for Authors
This review is for a review manuscript entitled: “Underexploited Marine Organisms” submitted to the journal Marine Drugs (Manuscript ID: marinedrugs-3709272). The manuscript describes products of marine organisms that are used in the cosmetics industry. Compounds derived from algae, microalgae and marine animals. Therefore, this work fits into the purpose of the journal Marine Drugs. The review covers 71 references from 1956 to 2025, most from the last 10 years. The text is clearly written. The introduction is somewhat poetic, but good. In summary, I found the work interesting. Therefore, the work may be considered for publication. Nevertheless, significant changes should be made in the revised version. Here are the following remarks:
Major comments.
The main point is the title-abstract-conclusion part of the manuscript. Strictly speaking, they are poorly informative. Usually, when someone reads the title, then the abstract, and finally the conclusion, they get a basic idea of ​​what the main topic of the paper is. Then you can read the main text to get more details. In a submitted paper, this does not work.
The title "Underexploited marine organisms" has a very broad meaning. It should be shown that it concerns cosmetics.
The abstract must be larger and show information about all parts of the paper.
The conclusion must summarize the findings presented in the paper as precisely as possible.
Minor comments. Editing errors.
Ginkgo biloba (line 90), it is not a substance
Furcellaria lumbricalis (italic) 102
Dilsea carnosa (italic) 125
References include numeration starting from number 27
Patent 35 is not properly cited
Figure 2. Diphlorethohydroxycarmalol formula. I do not understand why this compound is depicted but other discussed not (i.e. ishigoshide)
Author Response
The main point is the title-abstract-conclusion part of the manuscript. Strictly speaking, they are poorly informative. Usually, when someone reads the title, then the abstract, and finally the conclusion, they get a basic idea of ​​what the main topic of the paper is. Then you can read the main text to get more details. In a submitted paper, this does not work.
The title "Underexploited marine organisms" has a very broad meaning. It should be shown that it concerns cosmetics.
The abstract must be larger and show information about all parts of the paper.
The conclusion must summarize the findings presented in the paper as precisely as possible.
1) Title, summary and conclusion have been modified.
Ginkgo biloba (line 90), it is not a substance
2) It should be noted that no assertion was made that Ginkgo biloba was a substance in and of itself; the focus was instead on an extract of Ginkgo biloba.
Furcellaria lumbricalis (italic) 102
Dilsea carnosa (italic) 125
3) It should be noted that all the names of the organisms mentioned are now printed in italics.
References include numeration starting from number 27
4) Revised.
Patent 35 is not properly cited
5) The manner in which the patent is disclosed in the list of bibliographic references has undergone modification.
Figure 2. Diphlorethohydroxycarmalol formula. I do not understand why this compound is depicted but other discussed not (i.e. ishigoshide)
6) The decision has been taken to exclusively provide the formula for diphlorethohydroxycarmalol.
Round 2
Reviewer 1 Report
Comments and Suggestions for Authors
This paper has been improved well. I suggested the acceptance of this form.
Author Response
Thank you for your suggestions.
Reviewer 2 Report
Comments and Suggestions for Authors
The authors still did not address two major point raised by this reviewer:
1) The Introduction is too short, mainly focused on classic literature and not the scientific one, not really describing the potential of exploitation of bioactive molecules from marine organisms. Furthermore, it does not describe state of the art of the most used marine organisms for cosmetic compounds.
2) The “Conclusion” paragraph is barely there. In this last paragraph the authors should summarize comprehensively the results of their meta-analysis and give tentative predictions on future exploitability of the most promising organisms.
Author Response
We have revised the Introductions and Conclusions according to your suggestions.
Reviewer 3 Report
Comments and Suggestions for Authors
The review article (manuscript ID: marinedrugs-3709272) has been revised by the Authors. The comments presented in my review have been incorporated into the revised version of the manuscript. In particular, the title has been changed to the most precise one, as suggested. Also, section 2.3. Underexploited extracts of marine animals has been significantly revised. The conclusion has also been revised, although I expected a slightly different modification. Nevertheless, this is the Authors' concept. In my opinion, the manuscript can be considered for publication.
Author Response
Thank you for your suggestions.